# Plant Protection against Viruses: An Integrated Review of Plant Immunity Agents

**DOI:** 10.3390/ijms24054453

**Published:** 2023-02-23

**Authors:** Min Huang, Zilin Wu, Jingxin Li, Yuyu Ding, Shilin Chen, Xiangyang Li

**Affiliations:** 1State Key Laboratory Breeding Base of Green Pesticide and Agricultural Bioengineering, Key Laboratory of Green Pesticide and Agricultural Bioengineering, Ministry of Education, Guizhou University, Guiyang 550025, China; 2Faculty of Pharmaceutical Science, Guizhou University, Guiyang 550025, China

**Keywords:** plant protection, plant virus, plant immunity agents, molecular mechanism, review

## Abstract

Plant viruses are an important class of pathogens that seriously affect plant growth and harm crop production. Viruses are simple in structure but complex in mutation and have thus always posed a continuous threat to agricultural development. Low resistance and eco-friendliness are important features of green pesticides. Plant immunity agents can enhance the resilience of the immune system by activating plants to regulate their metabolism. Therefore, plant immune agents are of great importance in pesticide science. In this paper, we review plant immunity agents, such as ningnanmycin, vanisulfane, dufulin, cytosinpeptidemycin, and oligosaccharins, and their antiviral molecular mechanisms and discuss the antiviral applications and development of plant immunity agents. Plant immunity agents can trigger defense responses and confer disease resistance to plants, and the development trends and application prospects of plant immunity agents in plant protection are analyzed in depth.

## 1. Introduction

Currently, most conventional pesticides are expected to comprehensively prevent and control plant diseases by rapidly killing plant pathogens. However, the effect of host plant resistance to pathogens is ignored [1]. Plants have an immune defense system that is similar to that of higher animals, which has an important regulatory role in plant defense against pests, etc. The concept of a plant immune system was first introduced by American scientists in the journal *Nature* in 2006 [2]. In a 2007 study in the journal *Science*, German scientists reported that plants have special immune sensors that recognize microorganisms, viruses, and molds [3]. In recent years, with the development of science and with the advancement of technology, scientists have gradually determined the mechanisms related to plant immunity, which can produce an immune response by adjusting the defense system and metabolism when the plant is stimulated or in unfavorable conditions [4]. This defense response or immune response of plants can delay or mitigate the occurrence and development of plant disease, reduce the use of chemical pesticides and fertilizers, and reduce the residues resulting from agricultural products [5]. Plant immunity is usually induced by exogenous excitons, and scientists have developed excitons that can activate plant immunity into plant immunity-inducing resistance according to the immunity-inducing resistance characteristics of plants [6], and the immunity-inducing agents, in addition to comprehensive control, can also prevent frost, increase yield, and improve quality. They represent an effective way to solve environmental pollution, increase production safety, and achieve zero growth of pesticides.

Plant viruses represent one of the major crop pathogens, and although viral structures are simple, mutations are complex and have a large host range, posing a serious threat to crops and causing huge economic losses [7,8,9]. After virus infestation of plants, it interacts with host factors and causes changes in the physiological and biochemical characteristics of the plant, including the production of volatile substances, secondary metabolites, and other changes [10], leading to changes in the nutrient, metabolic, and signaling pathways of the plant, resulting in symptoms, such as localized greening and yellowing of the plant [11,12], ultimately affecting the growth and development of the plant. At present, methods to prevent and control viral damage are very limited, and prevention and control are mainly achieved with the help of insecticides to eliminate virus-spreading insects, which have developed resistance due to the misuse and overuse of insecticides, resulting in a significant environmental impact [13,14]. Although plants can generate antiviral responses through innate immunity, viruses are also evolving, leading to the loss of host resistance mechanisms, and genome-editing technologies can then be applied to antiviruses by viral diagnosis, resulting in the detection of viral genetic information and cutting viral genomes or altering plant genomes to enhance the innate immunity of plants [15,16], but the application of gene editing technologies is still very limited and has not been applied to a wide range of plants and field applications on a large scale. It is imperative to enhance host resistance to plant viruses by activating specific signaling cascades in order to induce systemic resistance and secondary metabolites in virus-infected plants to activate immune responses in the plant and enhance its resistance to viruses. Therefore, research on the generation and use of plant immunity agents has become a hot topic in the prevention and management of environmentally friendly plant disease. In 2015, the Chinese Ministry of Agriculture formulated an action plan to achieve zero growth of pesticides by 2020, stating that a resource-saving, environmentally friendly, sustainable pest and disease management technology system should be established. The research and development into pesticide reduction technology represents one of the key development technologies. The use of plant immuno-attractants to control plant viruses is a new idea and a new way to achieve plant protection. 

This review starts by explaining the principle of action of plant immunity agents, introduces plant immunity agents that are resistant to plant viruses, and summarizes their current applications in production practice in order to provide a scientific basis for the development and application of plant immunity agents to ensure future diversity.

## 2. Mechanisms of Plant Innate Immunity-Mediated Antiviral Resistance

Plants are attacked by a variety of pathogens, such as viruses, bacteria, fungi, nematodes, etc., during their growth and development. Plant viruses are one type of the crop pathogen. Although plant viruses are simple in structure, they are complex in molecular variation and pose a great threat to plant growth and development and agricultural production, resulting in huge economic losses. In the face of virus stress, plants use a variety of defense mechanisms to limit virus infestation, replication, and movement, with immune receptor signaling being among the more conservative mechanisms. The innate immunity stimulated by plant recognition of pathogenic microorganisms consists of two levels: specific recognition mediated by disease resistance genes (*R* genes) and broad-spectrum recognition mediated by cell surface receptors [17,18].

### 2.1. Disease Resistance Gene-Mediated Immunity

Disease *R* genes encode disease-resistant proteins, and the tobacco *N* gene was the first *R* gene identified to have some resistance to tobacco mosaic virus (TMV) [19,20]. At present, a large number of plant viral *R* genes have been cloned and identified, such as the *sw-1a*, *sw-1b*, *sw-2*, *sw-3*, *sw-4*, *sw-5*, *sw-5b*, *sw-6*, *sw-7*, *Sl5R-1*, and *Tsw* genes for defense against tomato spotted wilt virus (TSWV) in tomato [21,22,23,24,25]; *Ty-1*, *Ty-2*, *Ty-3*, *Ty-4*, *Ty-5*, and *Ty-6* for defense against tomato yellow leaf curl virus (TYLCV) [26]; the *Rx1* and *Rx2* genes for defense against potato virus X (PVX) in potato [27,28]; and tomato genes *Tm-1*, *Tm-2^2^*, which encode the proteins that interact with the viral replicase that interacts with viral genome replication, resulting in significant resistance to TMV [29,30]. Viral effectors compete with plant intracellular immune receptor proteins and evolve synergistically [2], and over the course of continuous evolution, pathogenic microorganisms such as viruses mutate or generate new effectors to escape or suppress effector-triggered immunity (ETI) in plants. New *R* genes also evolve in plants, and these new *R* genes reactivate ETI, causing the host to develop a disease resistance response, resulting in the plant having a hypersensitive response (HR) that prevents further virus infestation. 

### 2.2. Cell Surface Receptor-Mediated Immunity

The pattern recognition receptor (PRR) on the surface of plant cell membranes was also found to limit viral infection and to induce pathogen-associated molecular pattern (PAMP)-triggered immunity (PTI). The immune co-receptors BAK1 and BAK1-like kinase 1 (BKK1) contribute to the resistance of Arabidopsis to a variety of RNA viruses [31,32]. NSP-interacting kinase (NIK) protects against DNA viruses [33,34], and further studies have revealed that NIK1, when bound to ligands, enables the downstream component ribosomal protein L10 (RPL10) to translocate to the nucleus, while RPL10 is associated with the L10-interacting MYB domain-containing protein (LIMYB), which comprehensively represses the transcription of genes encoding ribosomal proteins, thereby systematically inhibiting translation and attenuating viral protein production and enhancing plant tolerance to viruses [35]. NIK1 synergizes with the LIMYB protein to inhibit systemic translation as a newly discovered defense mechanism against plant viruses, confirming that the PTI signaling pathway can also act against plant viruses. In addition, virus–plant interactions can modulate phytohormonal pathways and induce hormone-mediated defense responses, such as salicylic acid (SA) signaling, jasmonic acid (JA) signaling, and oleuropein iactone signaling [36,37,38,39,40], where different defense pathways cross each other and act synergistically.

### 2.3. SAR-Mediated Immunity

Systemic acquired resistance (SAR) is an immune response that is established under the control of two key immunologically active small metabolites: N-hydroxy tartrate (NHP) and SA [41]. SAR can be effective against subsequent infection [40,41]. SAR can be induced in distal tissues by the signaling compounds that are produced as mobile signals in pathogen-infected leaves by glycerol 3-phosphate (G3P), pipecolic acid (Pip), azelaic acid (AZA), NHP, nicotinamide adenine dinucleotide (NAD), nicotinamide adenine dinucleotide phosphate (NADP), and dehydrofolate (DA) [42,43,44,45,46,47]. During SAR activation, the levels of Pip and NHP increase in pathogen-infected and distal leaves, while the accumulation of Pip and NHP in systemic tissues coordinates the establishment of systemic immunity [41]. It is thus clear that SAR is an immune response of plants towards secondary infection by pathogens and is well worth studying because of its sustainable and long-term capacity in plant protection [48].

## 3. Plant Immunity Agents

Plant immunity agents can enhance the resilience of the immune system by activating plants to resist the harm incurred from pathogenic microorganisms (Figure 1). To date, more than 20 pesticides with immunity-inducing, disease-inductive properties have been registered worldwide.

### 3.1. Ningnanmycin

Ningnanmycin (Figure 2A) is an antiviral agent isolated from *Strepcomces noursei var*-xichangensisn, and Han et al. showed that ningnanmycin stimulates the activity of enzymes associated with defense responses in tobacco [49]. Ningnanmycin can induce enhanced phenylalaninase (PAL) activity and promote the production of sufficient phenolic compounds, lignin, and other minor substances to improve plant disease resistance against infection by viral particles, thereby significantly reducing plant symptoms. Ningnanmycin can increase tobacco superoxide dismutase (SOD) and peroxidase (POD) activity and reduce the amount of reactive oxygen species (ROS), which cause cell damage. It is able to stimulate the activity of *β*-1,3 glucanase and POD in tobacco and enhance the disease resistance of the plant. RT-PCR revealed that ningnanmycin can upregulate the expression of the *NPR1* and *Jaz3* genes related to plant defense response. The *Jaz3* gene is associated with JAZ protein synthesis, and JA signaling is switched by JAZ. JAZ proteins bind to and regulate the activity of downstream transcription factors in the absence of JA. However, JAZ proteins are broken down in the presence of JA or its bioactive derivatives, releasing transcription factors for the production of genes needed for the stress response [50], and plant immune agents enhance plant resistance by mediating gene expression in the JA pathway, thereby modulating JA signaling; activating receptor-like kinase FLS2, RLK1, and mitogen-activated protein kinase kinase (MAPKK), calcium signaling genes, and phytohormone-responsive genes; and activating multiple defense pathways to induce systemic resistance to TMV in tobacco [51]. These suggest that ningnanmycin can induce systemic resistance to TMV in a TMV host by stimulating plant defense signaling pathways and thus inducing systemic resistance to TMV.

### 3.2. Dufulin

Dufulin (Figure 2B), a green antiviral drug of the α-aminophosphonate ester class, is the world’s first immune-induced antiviral agent. It has low toxicity and low residue levels and is environmental friendly to nontargeted organisms [52]. Dufulin is a novel antiviral agent that activates SAR in plant (Figure 3B) [1]. Plant SAR utilizes SA, a phytohormone involved in defense responses, as a signal to aid in the development of disease resistance. By stimulating synthesis and accumulation of phycocyanin, SA mediates the expression of *R* genes, thereby activating the HR response to infections, resulting in programmed death [53,54]. Interestingly, Chen et al. found that dufulin could enhance the HrBP1 level to activate the SA signaling pathway to induce antiviral responses in host plants [55]. Additionally, it can bind to viral proteins and can inhibit virus infestation in rice [56,57]. According to recent studies, dufulin can also enhance the resistance to salt stress in rice by activating the SA pathway [58]. These suggest that dufulin not only enhances plant resistance to viruses, but also to abiotic stress.

### 3.3. Vanisulfane

Vanisulfane (Xiangcaoliusuobingmi) is a newly developed antiviral small molecule pesticide that exhibits good therapeutic and protective activity against many plant viruses. Vanisulfane is a derivative that contains a disulfide structure (Figure 2C) and was originally synthesized with a vanillin backbone by Zhang et al. in 2017. Vanisulfane has good therapeutic and protective activities against cucumber mosaic virus (CMV) and potato virus Y (PVY) [59]. Shi et al. conducted a study on the biological activity and mechanism of action of vanisulfane and found that vanisulfane enhances the activity of POD, PAL, SOD, and catalase (CAT) as well as the expression of defense genes in pepper, whose defense pathway may be the activation of the abscisic acid (ABA) signaling pathway (Figure 3C) [60]. Later, Shi et al. discovered that vanisulfane increased the activity of defense enzymes; increased the chlorophyll, flavonoids, and total phenol contents in plants to scavenge harmful free radicals; and enhanced defense genes involved in starch and sucrose metabolism, photosynthesis, the MAPK signaling pathway, and the oxidative phosphorylation pathway to improve plant immunity to pepper mild mottle virus (PMMoV) [61]. Many results have been obtained with vanisulfane derivatives, and these compounds have been determined to have strong protective and therapeutic activities against viruses, such as PVY and TMV, according to bioactivity assays [62,63]. These derivatives can be used as lead compounds for the design of plant immunity agents. 

### 3.4. Methiadinil

Methiadinil (Figure 2G), a novel plant immunity agents registered in China, was developed by Nankai University and displays good control efficiency against rice blast, rice sheath blight, cucumber gray mold, cucumber anthracnose, and TMV. Methiadinil can boost the expression of PAL, polyphenol oxidase (PPO), and SOD in rice, cucumber, and tobacco. According to Liu et al.’s research [64], this increase in disease resistance-related enzyme activity could lead to an improvement in crop output. Additionally, the results of SDS-polyacrylamide gel electrophoresis showed that methiadinil could induce the production of specific PRs in rice, cucumber, and tobacco. Methiadinil’s mechanism of action involves triggering the immune system of the host plant, enhancing its capacity for defense.

### 3.5. Cytosinpeptidemycin

Cytosinpeptidemycin (Figure 2E) is a bioactive secondary metabolite produced by rare *Streptomyces* isolated from soil in Liaoning. Yu et al. demonstrated that cytosinpeptidemycin can upregulate the expression of defense proteins, such as PR-5, PR-10, and heat shock protein (Hsp protein) as well as the activity of POD, CAT, and SOD to enhance the resistance of rice host plants to restrict the spread of Southern rice black-streaked dwarf virus (SRBSDV) (Figure 3A) [65]. Using Seq-RNA, An et al. [66] found that cytosinpeptidemycin significantly upregulates the expression of tobacco protein kinase 1 (NPK1), which was previously shown to negatively regulate IAA-inducible factor, and that upregulation of NPK1 may lead to the downregulation of growth hormone (Aux) [67]. It was also able to inhibit TMV viral RNA and protein accumulation in tobacco BY-2 protoplasts, delaying and suppressing TMV infection in *Nicotiana benthamiana* in multiple ways. The combination of cytosinpeptidemycin and amino-oligosaccharides also triggers ROS production and induces the upregulation of multiple defense response genes, such as *PR-1*, *PR-5*, *FLS2*, and *Hsp70*, which also affects viral subcellular localization and viral capsid formation [68], effectively inhibiting viral spread. The upregulation of serine and threonine protein kinase SAPK7 by cytosinpeptidemycin induces the ABA response and can upregulate certain *R* genes [66].

### 3.6. Oligosaccharins

Oligosaccharins are a new type of biogenic pesticide with bioactivities that include the promotion plant growth as well as antibacterial, antifungal, and insecticide properties. Amino-oligosaccharins (Figure 2D) are also representative antiviral agents among oligosaccharins. According to Yang et al. [69], amino-oligosaccharins can increase the expression of plant defense mechanisms by upregulating the expression of PRs in rice. Additionally, gene ontology (GO) research has revealed that proteins with differential expression are primarily located around the cytoplasm and cytomembrane and that their primary functions include catalytic activity, binding, and biosynthetic processes (Figure 3D).

Atailing is a newly reported agricultural antibiotic that has been registered in China and that has its own intellectual property rights. Kulye et al. [70] found that atailing could upregulate the expression of *PR1-a*, *PR1-b*, *PDF1.2*, *NPR1*, and *MAPK* to enhance the formation and accumulation of defensive materials (callosum, phenols, and xylogen) before induing the increased expression of PAL, SOD, PPO, and POD in tobacco. PR-1 is primarily mediated by SA and is sometimes referred to as an SA marker of pathogen-infected plants [71,72,73,74,75]. The PR1 protein exhibits sterol-binding activity, sequestering pathogen-produced sterols to prevent them from growing. The *NPR1* gene encodes the NPR1 receptor, and the SA signaling recognized by its receptor NPR1 induces NPR1 depolymerization into the nucleus. The NPR1 receptor also activates disease process-related gene expression by forming a transcription factor with TGAs [76,77,78]. The NPR1 receptor activates the expression of disease process-related genes through the formation of complexes with transcription factors, thereby triggering the plant immune response. Additionally, the MAPK cascade is involved in sending the signals caused by SA. Mechanistically speaking, by transmitting signals via NPR receptors, the MAPK cascade is engaged in regulating the dynamics of morphophysiological and molecular responses. The transport of SA from the cell membrane to the nucleus is subsequently mediated by these downstream signaling molecules, which causes the activation of defense genes to encourage plant resistance, which helps to resist the invasion and development of pathogens and to mitigate and prevent the occurrence of disease.

### 3.7. Acibenzolar-S-methyl

Acibenzolar-S-methyl (Figure 2H) initiates plant defense mechanisms [79,80]. Acibenzolar-S-methyl appears to be a functional analog of SA that activates SAR-related responses [81,82]. The application of acibenzolar-S-methyl induces the synthesis of PR proteins. According to previous reports, acibenzolar-S-methyl treatment enhances the activity of *β*-1,3-glucanase and chitinase [83], which catalyzes the hydrolysis of chitin and *β*-1,3-glucanase in wheat [84] and rice [85].

N-cyanomethyl-2-choloisonicotiamide (NCI) and 2,6-dichloroisonicotinic acid (INA) are acibenzolar-S-methyl derivatives and are defined as plant activators that display better control efficiency against cucumber fusarium wilt, Chinese cabbage clubroot, and sendai blight. It has been previously reported that INA and NCI can strengthen the expression of the genes *lox2*, *pr3*, *coi1*, and *pdf1.2*, activating plant disease resistance by activating the JA signaling pathway and thus enhancing plant resistance to RBSDV [86,87].

### 3.8. Phytohormone Abscisic Acid

Phytohormone abscisic acid (ABA) (Figure 2F) is known to play a vital role in the adaptive response of plants to various stresses, including drought, salt, and cold. ABA has two optical isomers, *S*-ABA and *R*-ABA, with *S*-ABA being the natural active form in plants. The occurrence of plant disease and insect pests is inevitable during the whole plant growth stage, and when plants are endangered by diseases, *S*-ABA can activate the *PIN* gene in plant leaf cells to produce protease inhibitors (flavonoids, quinones) and to prevent further damage to pathogens, thereby reducing the harm to plants [88]. In response to potato spindle tuber viroid (PSTVd) infection, ABA-related genes show different expression patterns in tomato cultivars, with some genes in the ABA biosynthetic pathway being upregulated and a few components of the guard cell ABA signaling pathway being downregulated. ABA is an active player in plant immunity against viruses and has a significant impact on plant defense against various pathogens [89]. According to reports, ABA functions as a defense against viruses through the callus, and it has been demonstrated that ABA inhibits the production of PR2, allowing more callus to accumulate in the and preventing viral infection [90]. It has been demonstrated that ABA causes callus buildup in several tissues and organelles by inhibiting *β*-1,3 glucanase (cell wall, sieve plate of the silique). However, RNA silencing is another defense mechanism mediated by ABA, and data indicate that ABA-dependent resistance against PVX and the Bamboo mosaic virus (BaMV) in Arabidopsis is mostly accomplished through this channel rather than through callus deposition [91,92]. If ABA is applied, it may affect the expression of other phytohormones. It will also suppress SA. For example, plant infestation by BaMV and CMV will result in the upregulation of ABA and in the suppression of SA [93], but the antiviral mechanism of this antagonistic effect remains to be investigated.

### 3.9. Vitamin C

Vitamin C (Vc) (Figure 3I) has been developed and registered as a plant immune agent in China. This product has been widely used in rice, tobacco, fruit, vegetables, tea, and other crops. It has been discovered that a Vc immune agents can robustly promote rice plants, lengthen the vegetative period, and be beneficial to the formation and accumulation of dry matter. Meanwhile, Vc immune agents promote growth and improve the immunity of wheat, significantly reduce the incidence of potato late blight, significantly enhance the disease resistance of pepper, and improve the disease resistance and quality of tea [94].

## 4. Conclusions and Perspectives

Plant immunity agents have multiple functions in disease resistance and are able to increase yield. In addition, they activate the molecular immune system in plants to improve disease resistance while also stimulating a series of metabolic regulatory systems in plants, which have multifunctional effects in promoting plant root and leaf growth, as well as the chlorophyll content, and in improving crop yield and stress resistance (low temperature, frost, drought and insect pests) [95,96,97,98]. The use of simple and effective viral detection methods to detect viral pathogenicity genes and the interaction of plant immunity agents and plant viral genes represents a direction for future research. Current viral diagnostic methods include CRISPRCas9, high-throughput sequencing (HTS), CRISPRCas-12a, and CRISPRCas-13a/d systems [16,99,100]. However, the complexity of the genetic variation observed in plant viruses requires further study. Due to the biological characteristics of plant viruses, it is difficult for conventional antiviral agents to kill viruses directly. The synergy of these defense strategies is expected to ensure a more robust and durable defense response against viruses.

Plant immunity agents, a new concept in vaccine engineering technology following human and animal vaccines, represent the new practice for the scientific control of disease and insect pests on the basis of the theory of the relationship among plants, pests, and biological pesticides and is the world’s most popular research field related to biological pesticide creation. Formulating the activation mechanism of plant immunity agents is the core scientific problem to improve disease resistance. We believe that, in the future, plant immunity agents will represent a new strategic industry with great development prospects.

## Figures and Tables

**Figure 1 ijms-24-04453-f001:**
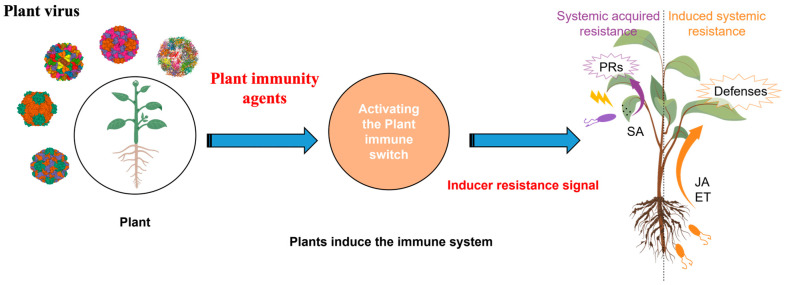
The main mechanism of plant immunity-induced resistance pattern is that once plant immunity agents are applied to crops, plant disease and insect pests are controlled by inducing crops to produce substances that resist or control plant disease and insect pests.

**Figure 2 ijms-24-04453-f002:**
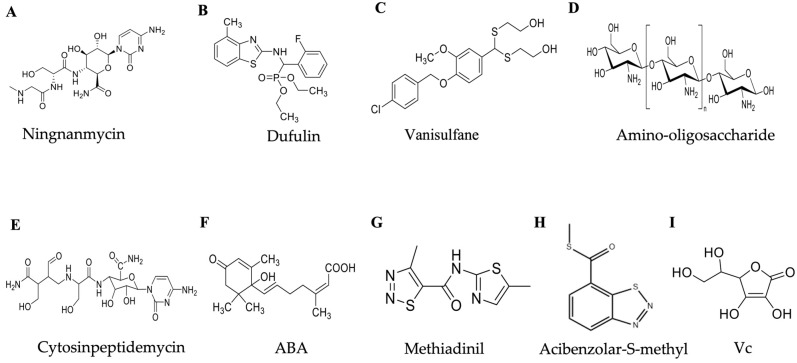
Structural formulas of immunity inducible agents. (**A**) Ningnanmycin. (**B**) Dufulin. (**C**) Vanisulfane. (**D**) Amino-oligosaccharide. (**E**) Cytosinpeptidemycin. (**F**) ABA. (**G**) Methiadinil. (**H**) Acibenzolar-S-methyl. (**I**) Vc.

**Figure 3 ijms-24-04453-f003:**
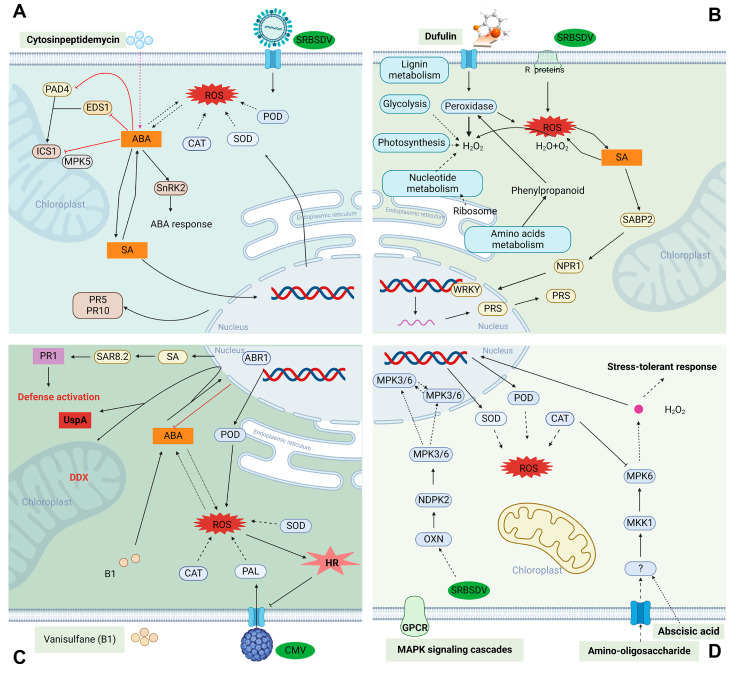
The action mechanism of plant immunity agents. (**A**) Cytosinpeptidemycin induces a significant upregulation of the expression of plant-related defense proteins (PR5, PR10, POD, SOD) and activates the host defense response. At the same time, cytosinpeptidemycin activates ABA molecular signaling and upregulates the expression of SnRK2 protein, which expresses ABA signaling activation, to produce an ABA response in the plant and inhibit the cellular damage caused by high SA concentrations through the ABA pathway that regulates SA concentration. (**B**) Dufulin induces the expression of SAR-related proteins such as POD, SA binding protein (SABP2), disease-process PRs, and benzoyltransferase, while toxafluorophos can bind to benzoyltransferase to enhance its activity and produce phycocyanin, combining with POD protein to trigger the host’s own immune system and thus enhancing the resistance of rice to SRBSDV. (**C**) Vanisulfane (B1) triggers the ABA pathway in peppers infected with cucumber mosaic virus. Enhancing the defensive enzyme activities of POD, PAL, SOD, and CAT; increasing the UspA content; promoting ABA biosynthesis; reducing SA accumulation; and ROS production. (**D**) Amino-oligosaccharide treatment induces specific upregulation of CAT expression through a phosphorylated protein cascade reaction. The high expression of CAT inhibits the production of H_2_O_2_, which ultimately leads to the elimination of harmful effects such as reactive oxygen species and the development of a stress tolerance response, resulting in increased plant resistance.

## Data Availability

Data sharing not applicable. No new data were created or analyzed in this study. Data sharing is not applicable to this article.

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
