# Peer review of "Plant Protection against Viruses: An Integrated Review of Plant Immunity Agents"

_ijms, 2023, doi:10.3390/ijms24054453_

Round 1

Reviewer 1 Report

This review paper summarized small molecules of inducing plant antiviral immunity. In general, it is a timing review although signficant improvement are needed.

Major comments:

1, Delete 2. plant virus (lines 41-196) I don't think it is necessary to give an introduction to these viruses. Instead, insert some related context when introducing the plant immunity inducible agents. 

2, Write a section about plant innate immunity and plant antiviral responses as the new Section 2.

3, Section 4 (Molecular mechanism) should be combined into Section 3 but not into a seperate section.

4, The language needs to be signficantly improved.

Minor comments:

1, plant immunity inducible agents, but not plant immunity agents.

2, rephrase line 29: ... and reduce the residue of agricultural products.

3, line 50, TMV is a member of the genus Tobamovirus of the family Virgaviridae.

4, line 51, ... broad host range and ready mechanical transmission.

5, line 67, ...immune inducers can trigger...

Author Response

Dear reviewer,

On behalf of my co-authors, we thank you very much for giving us an opportunity to revise our manuscript and we appreciate you very much for your positive and constructive comments and suggestions on our manuscript, we reply to your review comments in the attachment.

Author Response:

Dear Reviewer,

On behalf of my co-authors, we thank you very much for giving us an opportunity to revise our manuscript and we appreciate you very much for your positive and constructive comments and suggestions on our manuscript.

After we have carefully read the comments, we realized that the major merits of our work were not fully identified or recognized. The manuscript has been modified carefully following the reviewers’ comments. Each comment has been taken care of and answered point by point by referring to the relevant part of the manuscript. All the changes were based on the previous document and highlighted in “red color” in the revised version. We hope that the correction will meet with approval. Our point-by-point responses to reviewers’ comments.

Reviewer Comments:

Reviewer: 1

This review paper summarized small molecules of inducing plant antiviral immunity. In general, it is a timing review although significant improvement is needed.

  • Delete 2. plant virus (lines 41-196) I don't think it is necessary to give an introduction to these viruses. Instead, insert some related context when introducing the plant immunity inducible agents. Write a section about plant innate immunity and plant antiviral responses as the new Section 2.

Ans.: Thanks for your valuable comments and suggestions. In the revised version, the introduction of plant viruses has been removed and the content of plant innate immunity-mediated antiviral resistance has been recapitulated as the new Section 2.

  1. Section 4 (Molecular mechanism) should be combined into Section 3 but not into a seperate section.

Ans.: Thanks for your valuable comments and suggestions. The language of the whole manuscript has been carefully polished by MDPI English Editing in the revised version, and we attached the English Editing File, ID: English-58384.

  1. Plant immunity inducible agents, but not plant immunity agents.

Ans.: Thanks for your valuable comments and suggestions. “Plant immunity agents” was replaced by “      Plant immunity inducible agents” in the revised version.

  1. Rephrase line 29: ... and reduce the residue of agricultural products.

Ans.: Thanks for your valuable comments and suggestions. The content of this section you mentioned has been removed and the content you suggested has been written in the revised version.

  1. Line 50, TMV is a member of the genus Tobamovirus of the family Virgaviridae.

Ans.: Thanks for your valuable comments and suggestions. The content of this section you mentioned has been removed and the content you suggested has been written in the revised version. We rewrote that TMV is a member of the genus Tobamovirus of the family Virgaviridae.

  1. Line 51, ... broad host range and ready mechanical transmission.

Ans.: Thanks for your valuable comments and suggestions. The content of this section you mentioned has been removed and the content you suggested has been written in the revised version.

  1. Line 67, ...immune inducers can trigger.

Ans.: Thanks for your valuable comments and suggestions. The content of this section you mentioned has been removed and the content you suggested has been written in the revised version.

Thank you and best regards.

Prof. & Xiangyang Li

State Key Laboratory Breeding Base of Green Pesticide and Agricultural Bioengineering, Key Laboratory of Green Pesticide and Agricultural Bioengineering, Ministry of Education, Guizhou University, Huaxi District, Guiyang 550025, PR China.

E-mail: [email protected] (X.Y.L.).

Reviewer 2 Report

Please find the attachemnt with the corrected version of the manuscript. Sentences which need special attention due to necessary corrections are label yellow.

The manuscript "Plant Protection against Virus..." needs extensive English correction begining from the title to the last sentence of Conclusion.

Unfotunately, this manuscript is very chaotic, and it is difficult to understand what Authors wants to show to Redears about this important topic as modulation of plant innate immunity to enhanced resistance to viruses. The manuscript should be re-written and focused on important and selected aspects of plant immunity inducers against viruses.  In present form, the manuscript is a mixture of mostly general information on plant responses to pathogens which are also activated against viruses. The manuscript lacks  any novel approach in the description of plant protection against viral diseases.

Author Response

Dear reviewer,

On behalf of my co-authors, we thank you very much for giving us an opportunity to revise our manuscript and we appreciate you very much for your positive and constructive comments and suggestions on our manuscript, we reply to your review comments in the attachment.

Dear Reviewer,

On behalf of my co-authors, we thank you very much for giving us an opportunity to revise our manuscript and we appreciate you very much for your positive and constructive comments and suggestions on our manuscript.

After we have carefully read the comments, we realized that the major merits of our work were not fully identified or recognized. The manuscript has been modified carefully following the reviewers’ comments. Each comment has been taken care of and answered point by point by referring to the relevant part of the manuscript. All the changes were based on the previous document and highlighted in “red color” in the revised version. We hope that the correction will meet with approval. Our point-by-point responses to reviewers’ comments.

Reviewer Comments:

Reviewer: 2

Please find the attachment with the corrected version of the manuscript. Sentences which need special attention due to necessary corrections are label yellow.

The manuscript "Plant Protection against Virus..." needs extensive English correction beginning from the title to the last sentence of Conclusion.

Unfortunately, this manuscript is very chaotic, and it is difficult to understand what Authors wants to show to Readers about this important topic as modulation of plant innate immunity to enhanced resistance to viruses. The manuscript should be re-written and focused on important and selected aspects of plant immunity inducers against viruses.  In present form, the manuscript is a mixture of mostly general information on plant responses to pathogens which are also activated against viruses. The manuscript lacks  any novel approach in the description of plant protection against viral diseases.

Ans.: Thanks for your valuable comments and suggestions. Based on the suggested changes in the yellow section marked in your attachment, we have revised it in the revised version.

  • The title has been changed to “Plant Protection against Viruses of Plant Immunity Inducible Agents”.
  • Figure 3 illustrates the mechanism based on the literatures (A: Yu, L et al. Label-free quantitative proteomics analysis of Cytosinpeptidemycin responses in Southern rice black-streaked dwarf virus-infected rice. Biochem. Physiol. 2018, 147, 20-26ï¼›B: Song, B. A et al. Innovation and application of environment-friendly antiviral agents for plants. Springier. Berlin. 2009, 21, 207-300; C: Shi, J et al. Proteomics analysis of Xiangcaoliusuobingmi-treated Capsicum annuum L. infected with Cucumber mosaic virus. Pestic. Biochem. Physiol. 2018, 149, 113-122; D: Yang, A et al. Label-free quantitative proteomic analysis of chitosan oligosaccharide-treated rice infected with Southern rice black-streaked dwarf virus. Viruses, 2017, 9, 115.).
  • In line 270, the words "secondarymetabolites" have been changed to "secondary metabolites".
  • While we reworked and combed the article to focus on the important and selected aspects of plant immune inducers against viruses.
  • The language of whole manuscript has been carefully polished by MDPI English Editing in the revised version, and we attached the English Editing File, ID: English-58384.

Thank you and best regards.

Prof. & Xiangyang Li

State Key Laboratory Breeding Base of Green Pesticide and Agricultural Bioengineering, Key Laboratory of Green Pesticide and Agricultural Bioengineering, Ministry of Education, Guizhou University, Huaxi District, Guiyang 550025, PR China.

E-mail: [email protected] (X.Y.L.).

Reviewer 3 Report

The abstract is missing key information on necessity of the review. It needs to be revised and can be elaborated.

The introduction needs to broadened to consider alternative approaches such as CRISPR-Cas to improve resistance against plant. I suggest authors to use following reference for the discussion: Hinge et al. Engineering Resistance Against Viruses in Field Crops Using CRISPR- Cas9. Curr Genomics. 2021 Oct 18;22(3):214-231. doi: 10.2174/1389202922666210412102214.

Figure 2: the resolution needs to be improved.

A conclusions section can added, but not necessary.

Figure 4: could be enlarged for better appeal. Also increase the font size to make the legends clearly visible.

A brief discussion on plant virus diagnostics is encouraged, for example, Chavhan et al. Sequence analysis of coat protein and molecular profiling of sunflower necrosis virus (SNV) strains from Indian subcontinent. J. Plant Biochem. Biotechnol. 27, 28–35 (2018). https://doi.org/10.1007/s13562-017-0412-z.

The review could be considered after careful review as per above suggestions.

Author Response

Dear reviewer,

On behalf of my co-authors, we thank you very much for giving us an opportunity to revise our manuscript and we appreciate you very much for your positive and constructive comments and suggestions on our manuscript, we reply to your review comments in the attachment.

Author Response: 

Dear Reviewer,

On behalf of my co-authors, we thank you very much for giving us an opportunity to revise our manuscript and we appreciate you very much for your positive and constructive comments and suggestions on our manuscript.

After we have carefully read the comments, we realized that the major merits of our work were not fully identified or recognized. The manuscript has been modified carefully following the reviewers’ comments. Each comment has been taken care of and answered point by point by referring to the relevant part of the manuscript. All the changes were based on the previous document and highlighted in “red color” in the revised version. We hope that the correction will meet with approval. Our point-by-point responses to reviewers’ comments.

Reviewer Comments:

Reviewer: 3

  1. The abstract is missing key information on necessity of the review. It needs to be revised and can be elaborated.

Ans.: Thanks for your valuable comments and suggestions. The content of the abstract has been added with key information in the revised version.

  1. The introduction needs to broaden to consider alternative approaches such as CRISPR-Cas to improve resistance against plant. I suggest authors to use following reference for the discussion: Hinge et al. Engineering Resistance Against Viruses in Field Crops Using CRISPR-Cas9. Curr Genomics. 2021 Oct 18;22(3):214-231. doi: 10.2174/1389202922666210412102214.

Ans.: Thanks for your valuable comments and suggestions. The introduction has been broadened according to the literature as suggested.

  1. Figure 2: the resolution needs to be improved.

Ans.: Thanks for your valuable comments and suggestions. The resolution of the image has been increased after the modification.

  1. A conclusions section can add, but not necessary.

Ans.: Thanks for your valuable comments and suggestions. The conclusions and perspectives are placed in section 4 in the revised version.

  1. Figure 4: could be enlarged for better appeal. Also increase the font size to make the legends clearly visible.

Ans.: Thanks for your valuable comments and suggestions. The content related to images has been improved in the revised version

  1. A brief discussion on plant virus diagnostics is encouraged, for example, Chavhan et al. Sequence analysis of coat protein and molecular profiling of sunflower necrosis virus (SNV) strains from Indian subcontinent. J. Plant Biochem. Biotechnol. 27, 28–35 (2018). https://doi.org/10.1007/s13562-017-0412-z.

Ans.: Thanks for your valuable comments and suggestions. Plant virus diagnosis is discussed in lines 315-322 in the revised version.

Thank you and best regards.

Prof. & Xiangyang Li

State Key Laboratory Breeding Base of Green Pesticide and Agricultural Bioengineering, Key Laboratory of Green Pesticide and Agricultural Bioengineering, Ministry of Education, Guizhou University, Huaxi District, Guiyang 550025, PR China.

E-mail: [email protected] (X.Y.L.).

Round 2

Reviewer 1 Report

It is glad to see the improvements of the manuscript in the structure, content, and language. Therefore, I only have two minor comments for further improving the manuscript:

1, A section to introduce the systemic acquired resistance (SAR) is missing.

2, the manuscript can be carefully read thoroughly to further polish the language.

Author Response

Thank you again for your review and comments. We have answered your comments and suggestions in the attachment.

Author Response: 

Dear Reviewer,

On behalf of my co-authors, we thank you very much for giving us an opportunity to revise our manuscript and we appreciate you very much for your positive and constructive comments and suggestions on our manuscript.

After we have carefully read the comments, we realized that the major merits of our work were not fully identified or recognized. The manuscript has been modified carefully following the reviewers’ comments. Each comment has been taken care of and answered point by point by referring to the relevant part of the manuscript. All the changes were based on the previous document and highlighted in “red color” in the revised version. We hope that the correction will meet with approval. Our point-by-point responses to reviewers’ comments.

Reviewer Comments:

Reviewer: 1

It is glad to see the improvements of the manuscript in the structure, content, and language. Therefore, I only have two minor comments for further improving the manuscript:

  1. A section to introduce the systemic acquired resistance (SAR) is missing.

Ans.: Thanks for your careful review and comments again. The SAR introduction has been added in lines 116-128.

  1. The manuscript can be carefully read thoroughly to further polish the language.

Ans.: Thanks for your careful review and comments again. We have asked MDPI again to polish the manuscript.

Thank you and best regards.

Prof. & Xiangyang Li

State Key Laboratory Breeding Base of Green Pesticide and Agricultural Bioengineering, Key Laboratory of Green Pesticide and Agricultural Bioengineering, Ministry of Education, Guizhou University, Huaxi District, Guiyang 550025, PR China.

E-mail: [email protected] (X.Y.L.).

Reviewer 2 Report

Please find the attachment with minor issues taht need to be addressed.

Tha manuscript was sufficiently corrected.

Author Response

Thank you again for your review and comments. We have answered your comments and suggestions in the attachment.

Dear Reviewer,

On behalf of my co-authors, we thank you very much for giving us an opportunity to revise our manuscript and we appreciate you very much for your positive and constructive comments and suggestions on our manuscript.

After we have carefully read the comments, we realized that the major merits of our work were not fully identified or recognized. The manuscript has been modified carefully following the reviewers’ comments. Each comment has been taken care of and answered point by point by referring to the relevant part of the manuscript. All the changes were based on the previous document and highlighted in “red color” in the revised version. We hope that the correction will meet with approval. Our point-by-point responses to reviewers’ comments.

Reviewer Comments:

Reviewer: 2

Please find the attachment with minor issues that need to be addressed. The manuscript was sufficiently corrected.

Ans.: Thanks for your careful review and comments again. Based on the suggested changes in your attachment, we have revised it in the revised version.

  • “diseases” was replaced by “pathogens” in line 9。
  • In line 294, the words " avonoids" have been changed to " flavonoids".

Thank you and best regards.

Prof. & Xiangyang Li

State Key Laboratory Breeding Base of Green Pesticide and Agricultural Bioengineering, Key Laboratory of Green Pesticide and Agricultural Bioengineering, Ministry of Education, Guizhou University, Huaxi District, Guiyang 550025, PR China.

E-mail: [email protected] (X.Y.L.).

Reviewer 3 Report

Authors have not revised the manuscript as per suggestions.
No changes were highlighted.
Authors have not revised the manuscript as per suggestions. No changes were highlighted. Grammar need attention, for example, correct title should be: Plant Protection against Viruses: An Integrated Review of Plant Immunity Agents

Author Response

Thank you again for your review and comments. We have answered your comments and suggestions in the attachment.

Author Response: 

Dear Reviewer,

On behalf of my co-authors, we thank you very much for giving us an opportunity to revise our manuscript and we appreciate you very much for your positive and constructive comments and suggestions on our manuscript.

After we have carefully read the comments, we realized that the major merits of our work were not fully identified or recognized. The manuscript has been modified carefully following the reviewers’ comments. Each comment has been taken care of and answered point by point by referring to the relevant part of the manuscript. All the changes were based on the previous document and highlighted in “red color” in the revised version. We hope that the correction will meet with approval. Our point-by-point responses to reviewers’ comments.

Reviewer Comments:

Reviewer: 3

Authors have not revised the manuscript as per suggestions. No changes were highlighted. Grammar need attention, for example, correct title should be: Plant Protection against Viruses: An Integrated Review of Plant Immunity Agents 

Ans.: Thanks for your careful review and comments again. The major modifications are as follows:

  • As suggested by another reviewer, “plant Immunity Agents” was replaced by “plant immunity inducible agents”, therefore we have changed the title to “Plant Protection against Viruses: An Integrated Review of Plant Immunity Inducible Agents” in the revised version.
  • The font of the figures has been enlarged for a clearer resolution.
  • The language of the whole manuscript has been carefully polished by MDPI English Editing in the revised version, and we attached the English Editing File, ID: English-58384.

Thank you and best regards.

Prof. & Xiangyang Li

State Key Laboratory Breeding Base of Green Pesticide and Agricultural Bioengineering, Key Laboratory of Green Pesticide and Agricultural Bioengineering, Ministry of Education, Guizhou University, Huaxi District, Guiyang 550025, PR China.

E-mail: [email protected] (X.Y.L.).

Round 3

Reviewer 3 Report

Authors have not highlighted the changes during revision. 

I do not find any improvement in the text from previous drafts.

I can see there are several mistakes in English, including the title of  the manuscript is wrong. Correct title would be: "Plant Protection Against Viruses: An Integrated Review of Plant Immunity Agents"

Author Response

Author Response: 

Dear Reviewer,

On behalf of my co-authors, we thank you very much for giving us an opportunity to revise our manuscript and we appreciate you very much for your positive and constructive comments and suggestions on our manuscript.

After we have carefully read the comments, we realized that the major merits of our work were not fully identified or recognized. The manuscript has been modified carefully following the reviewers’ comments. Each comment has been taken care of and answered point by point by referring to the relevant part of the manuscript. All the changes were based on the previous document and highlighted in “red color” in the revised version. We hope that the correction will meet with approval. Our point-by-point responses to reviewers’ comments.

Reviewer Comments:

Reviewer: 3

1.Authors have not highlighted the changes during revision.

Ans.: Thanks for your careful review and comments again. We have previously highlighted the changes in red in the manuscript and placed this manuscript in the supplementary material. We apologize that you may not have seen it in this way, but after revising it again in response to your comments, we have also highlighted the changes in red and uploaded the manuscript into the submission system.

2.I do not find any improvement in the text from previous drafts.

Ans.: Thanks for your careful review and comments again. The manuscript has been revised as follows in accordance with the reviewers' comments.

  • When the manuscript was first submitted, section 2 of the manuscripts was about plant viruses, and this section was replaced by mechanisms of plant innate immunity-mediated antiviral resistance as suggested by several other reviewers.
  • The mechanisms of action in section 4 were combined with section 3 on plant immunity agents as suggested by several other reviewers.
  • Plant immunity and plant viruses have been added to the introduction section according to your suggestion in the revised version.
  • The resolution and content of the figures has been increased after the modification.
  • Plant virus diagnosis is discussed in lines 327-332 according to your suggestion in the revised version.

The language of the whole manuscript has been carefully polished by MDPI English Editing in the revised version, the English Editing File, ID: English-58384.

3.I can see there are several mistakes in English, including the title of  the manuscript is wrong. Correct title would be: "Plant Protection Against Viruses: An Integrated Review of Plant Immunity Agents"

Ans.: Thanks for your careful review and comments again. We carefully checked and revised it as follows.:

  • we have changed the title to “Plant Protection against Viruses: An Integrated Review of Plant Immunity Agents” in the revised version.
  • “virus” was replaced by “ viral“ in line 42.
  • “diseases” was replaced by “ disease“ in the revised version.
  • In line 85 “virus” was replaced by “ viral“.
  • Deleted “agent resistance” in line 155.
  • Deleted “and” in line 235.
  • In line 169 “environmentally friendly” was replaced by “ environmental friendliness “ .
  • In line 159 and line 178 “This” was replaced by “These”.
  • In line 225 “virus” was replaced by “ viral”.
  • In line 229 “are” was replaced by “ is“.
  • In line 315 “agent” was replaced by “agents ”.
  • In line “includeCRISPRCas9” was replaced by “include CRISPRCas9”.
  • “plant immunity inducible agents” was replaced by “plant immunity agents” in the revised version.

Thank you and best regards.

Prof. & Xiangyang Li

State Key Laboratory Breeding Base of Green Pesticide and Agricultural Bioengineering, Key Laboratory of Green Pesticide and Agricultural Bioengineering, Ministry of Education, Guizhou University, Huaxi District, Guiyang 550025, PR China.

E-mail: [email protected] (X.Y.L.).

Round 4

Reviewer 3 Report

Somehow I did not notice any changes in the manuscript including the the suggested title modifications. 

Author Response

Dear Reviewer,

On behalf of my co-authors, we thank you very much for giving us an opportunity to revise our manuscript and we appreciate you very much for your positive and constructive comments and suggestions on our manuscript.

After we have carefully read the comments, we realized that the major merits of our work were not fully identified or recognized. The manuscript has been modified carefully following the reviewers’ comments. Each comment has been taken care of and answered point by point by referring to the relevant part of the manuscript. All the changes were based on the previous document and highlighted in “red color” in the revised version. We hope that the correction will meet with approval. Our point-by-point responses to reviewers’ comments.

Reviewer: 3

  1. Somehow, I did not notice any changes in the manuscript including the the suggested title modifications.

Ans.: Thanks for your careful review and comments again. We have certainly revised the manuscript in accord with your comments and those of other reviewers, and the changes are highlighted in red, but it is really not clear what caused you not to see them. This time we have re-uploaded the revised manuscript for your review.

Thank you and best regards.

Prof. & Xiangyang Li

State Key Laboratory Breeding Base of Green Pesticide and Agricultural Bioengineering, Key Laboratory of Green Pesticide and Agricultural Bioengineering, Ministry of Education, Guizhou University, Huaxi District, Guiyang 550025, PR China.

E-mail: [email protected] (X.Y.L.).

Round 5

Reviewer 3 Report

Now I noticed the changes made by authors. I feel the manuscript is revised well. It can be considered for publication.